# Proposal of a Token-Based Node Selection Mechanism for Node Distribution of Mobility IoT Blockchain Nodes

**DOI:** 10.3390/s23198259

**Published:** 2023-10-05

**Authors:** Jinsu Kim, Eunsun Choi, Byung-Gyu Kim, Namje Park

**Affiliations:** 1Department of Convergence Information Security, Graduate School, Jeju National University, Jeju 63243, Republic of Korea; kimjinsu@jejunu.ac.kr; 2Creative Education Base Center, Jeju National University, Jeju 63294, Republic of Korea; choi910624@jejunu.ac.kr; 3Department of Artificial Intelligence Engineering, Sookmyung Women’s University, Seoul 04310, Republic of Korea; bg.kim@sookmyung.ac.kr; 4Department of Computer Education, Teachers College, Jeju National University, Jeju 63243, Republic of Korea

**Keywords:** blockchain, mobility, random selection, encoding, token

## Abstract

Various elements, such as evolutions in IoT services resulting from sensoring by vehicle parts and advances in small communication technology devices, have significantly impacted the mass spread of mobility services that are provided to users in need of limited resources. In particular, business models are progressing away from one-off costs towards longer-term costs, as represented by shared services utilizing kick-boards or bicycles and subscription services for vehicle software. Advances in shared mobility services, as described, are calling for solutions that can enhance the reliability of data aggregated by users leveraging mobility services in the next-generation mobility areas. However, the mining process to renew status ensures continued network communication, and block creation demands high performance in the public block chain. Therefore, easing the mining process for state updates in public blockchains is a way to alleviate the high-performance process requirements of public blockchains. The proposed mechanism assigns token-based block creation authority instead of the mining method, which provides block creation authority to nodes that provide many resources. Blocks are created only by a group of participants with tokens, and after creation, tokens are updated and delivered to new nodes to form a new token group. Additionally, tokens are updated in each block after their initial creation, making it difficult to disguise the tokens and preventing resource-centered centralization.

## 1. Introduction

Unlike internal combustion engine vehicles, today’s vehicles are becoming more electric, and this means increasing the electrification of various devices that are controlling driving by subjecting many vehicle parts to control by sensors [1,2]. On top of this, advances in IoT communication technology help personally-owned bicycles and kick-boards be remote controlled and protected against theft, which in turn is an accelerator for shared services. The mobility sector is currently focusing on building shared mobility while also conducting research in the area of meta-mobility, which combines virtual environments of the metaverse with mobile services, as the next generation of mobility. Meta-mobility aims to provide users with immersive experiences through various mobile services. In simple terms, research is being conducted on meta-mobility, which combines various mobile services and immersive experiences in the field while emphasizing the establishment of shared mobility [3,4,5].

Meta-mobility, which provides users with immersive services based on a virtual environment, can create a virtual space and provide an environment for users. In another way, smart devices can be used to directly photograph a location desired by the user and make it feel real. In order to provide users with a sense of presence in a specific place in real-time, a meta-mobility device that can inform the user of the situation in the field is required, and a lot of communication is required [6,7,8]. In particular, strengthening the integrity of communication is a very important factor, given the presence of physical devices. If the integrity of communication cannot be guaranteed, control of the physical device becomes impossible and may pose a risk. Blockchain can be an effective means in that it can strengthen the integrity of communication [9].

Block chain is apparently an attractive solution in that it guarantees the integrity of mobility data and helps contracts be implemented on a real-time basis. Yet, there are many challenges to applying it in a limited mobility environment. It guarantees transparent transactions and reinforces the integrity of block data as blocks increase, which makes it a good choice in many areas [10,11,12]. In particular, PoW, which does not restrict participants, faces more issues to be addressed in order to acquire compensation for creating transaction blocks. Mobility environments, in general, do not require high performance, and shared mobility devices such as bicycles and kick-boards should consider low performance and energy efficiency since they are based on the IoT environment. This requires PoW to perform computation in order to continuously obtain a hash value to compete for block creation. This also suggests limitations in applying in a shared mobility environment in that client-to-client communication has to be made [13,14,15,16].

Compared to PoW, which exhausts resources for hash computation to create blocks, PoS, which acquires block creation authority corresponding to one’s share, can substitute resources consumed for hash computation by proving its own share. This enables applications with lower performance levels. However, the fact that it acquires authority as per share and subsequent compensation means it is prone to fixation [17,18].

The mechanism proposed in this paper provides tokens to create blocks by selecting random nodes to reinforce data integrity in a limited scope whereby share from block creation does not affect mobility service in itself. In addition, the use of encrypted tokens in the course of sending them decodes encrypted tokens only at specific nodes, ensuring communication without disclosing selected nodes in the network. This paper analyzes the shared mobility network method in Section 2 and introduces the analysis of existing block network methods and research trends. Section 3 introduces a proposed mechanism for constructing a token-based block network. In Section 4, we analyze and introduce the differences from the existing block network method. Section 5 summarizes the overall content and introduces the proposed mechanism in summary.

## 2. Related Studies

### 2.1. Shared Mobility Service Network

There are many network standards that include NB-IoT (NarrowBand-Internet of Things), LoRa (Long Range), and eMTC (enhanced Machine-Type Communication) for application to mobility services. In particular, low power, low performance, and higher energy efficiency are in demand for use in such limited performance as IoT. The IoT network standard focuses more on energy efficiency and data transmission distance than on speed itself since it has to be applied in a limited IoT environment [19,20].

LoRa is a wireless communication technology for low power, long-distance communication. It derives from CSS (Chirp Spead Spectrum) technology. Its low transmission means it is applicable to IoT that sends small-sized data, and the transmission speed is 290 bps in 14 km coverage and 5470 bps in 2 km coverage. It consumes less battery since it transmits data and maintains a consistent time interval [21].

NB-IoT is the standard low power, broad communication technology established by the 3GPP (Third Generation Partnership Project) mobile communication standardization organization. With a narrow bandwidth of 180 kHz, it supports 250 kbps in data transmission speed and broad service beyond 10 km. It is suitable for fixed smart services since it does not support mobility. It is applicable to an IoT environment characterized by less frequent data use, low power consumption, and low mobility. It is also ideal for metering sensors such as water, gas, electricity, and temperature as well as smart lights and location-tracing devices that are located in remote areas [22].

eMTC, which is also known as LTE cat M1, was standardized by 3GPP’s Release 13. It provides up to 1 Mbps in data transmission speed at a 1.4 MHz bandwidth. It also provides VoLTE (Voice over Long Term Evolution). Its high data transmission speed makes it a good application for technologies that require heavy data, such as tracking mobile objects or real-time services [23].

As shown in Table 1, which compares IoT communication technologies, environments with less mobility do not require real-time processing from LoRa or NB-IoT technologies. Rather, they are applied to environments with smaller data transmission volumes. In mobility services, on the other hand, data has to be processed in real-time, and mobility has to be guaranteed, too. In mobility services that require mobility and real-time processing, such as kick-boards in Korea, eMTC is applied.

### 2.2. Blockchain Consensus Mechanism

Block chain consensus mechanisms include PoW (Proof of Work), PoS (Proof of Stake), DPoS (Delegated Proof of Stake), etc.

PoW, which is the most well-known mechanism, gives block creation authority to the node that used the largest amount of resources to creating the block chain’s block. The process of substituting the nonce value on a repeated basis to find the hash value below the target value, thereby solving the largest number of problems and having the the authority to create blocks, is called mining [24,25,26]. Bit coin, which is the most popular example of PoW, is huge in size, and its network is protected from attack by 51%, which is capable of exercising block’s forgery by participating node 51%, which is one of the weaknesses of block chain. This is because the economic value required to acquire the node’s 51% computing power to prevent data forgery is exponentially high and therefore poor in efficiency [27]. The difficulty of mining is set in a way to produce a certain interval as per the computation level of nodes taking part in the mining process, which subsequently, aggravated competition for mining using high-specification equipment. This, in turn, made mining more challenging and mining equipment that is higher in performance in need. Growing necessity of high-performance computing equipment is consuming more electric power and causing too much waste of energy [28,29].

PoS is an algorithm that grants decision-making authority to nodes based on shares. It take parts in creating blocks by proving its shares to the block and therefore does not demand resources consumed in the course of mining, which is unlike PoW, which competes to acquire compensation [30,31]. As such, it consumes less energy, and all nodes that have shares take part in decision-making since blocks’ updating authority is matched by shares. One key example operating in PoS is ethereum. Its consensus mechanism was based on PoW at first but has since been converted to PoS. Thanks to this conversion, mining not only consumes less electricity but also mitigates environmental issues such as heat generated during mining. These changes are not without downsides. Compensation in the PoS method makes distributions based on shares. This means higher compensation to nodes with higher shares, which subsequently has a bigger impact on the network and issues associated with coins concentrated in specific nodes [32,33].

The DPoS method is an algorithm whereby nodes can exercise their voting rights as per shares and engage in decision-making via the selected representative. Since it appoints a representative that will make decisions on behalf by acquiring voting rights per share, transaction approval by a few representatives can accelerate processing speed relative to a PoS solution where all nodes take part [34]. However, the fact that blocks are created only by a few selected proxies is a far departure from meeting the objective of block chaining, which is decentralization. Another downside is that participants with fewer shares are selected as representatives. Those with many shares in a DPoS environment do not want to see their shares undermined, and this can promote participants’ reliability. On the contrary, such participants are at higher risk of undermining shares, and this could adversely impact the block chain environment [35,36].

### 2.3. Related Research Trends

Various studies on mobility services leveraging block chains are in progress in sync with advances in shared mobility services. In particular, there are many studies underway to share vehicles and apply them in low-performance environments without limiting the scope of research to apply block chains in mobility environments based on IoT.

In her research, Sophia examined shared mobility environments based on IoT and particularly focused on cases applying to vehicle sharing and vehicle lease environments. The research proposed the architecture of a platform based on IoT and block chain to promote shared mobility by combining vehicle sharing and leasing and realized a platform that simplifies sharing and leasing procedures. In research by Madhusudan [37], it was mentioned that intelligent vehicles are recording innovative growth, but there are a lot of security vulnerabilities at the same time and challenges in safely sharing data with traditional ways of security protection. As a solution, the research suggested determining the security elements that are required for data sharing and sharing intelligent vehicle data. In another one by Madhusudan [38], he mentioned that intelligent vehicles perform vehicle-to-object communication based on the Internet, and such a communication environment brings with it a wide variety of security vulnerabilities. Major issues witnessed in intelligent vehicle communication include reception in communication channels, data reliability, accuracy, and security, and studies to build reliable intelligent transportation systems by applying block chain as solutions were carried out.

Research on shared mobility environments is not limited to vehicles. In fact, research is being made in various areas, such as bicycle sharing services. Hanyue identified leakage of users’ personal information by shared bicycles and property damage as key issues in his research [39]. As a solution to this, he proposed a bicycle sharing system based on a block chain service platform and a C2C (consumer-to-consumer) shared operation method to address the limitations of the existing centralized method. Daozhi’s research [40] looked into a system that taps into smart contracts to prevent issues arising from companies declining to return user deposits when they discontinue bicycle sharing services in the course of the rapid growth of the platform.

## 3. Proposal of a Block Network Random Authentication Node Selection Mechanism

The objective of this paper is to record information such as user information, mobility device information, and payment information in a shared mobility environment in block networks. The general mobility environment is defined by equipment put to use for services by companies. Hence, block networks in a mobility environment do not need to register countless users in the open network. Block networks, therefore, form a private block chain structure accessible only to certified users. Private block chains, however, can be modulated by a few upper nodes dominated in the process of forming transactions since authority over user participation and block creation is handled by an upper body. Hence, upper nodes go through an authentication process to access block networks in this paper, and block creation authority is performed by nodes in the block network. Block creation authority in the block network is performed by tokens, which are authentication means to acquire authority to create blocks. As a way to transmit random nodes on the block network after blocks are created it averts monopolies.

Figure 1 is the general concept of the mechanism proposed. Mobility devices are a concept covering kick-boards, bicycles, and intelligent vehicles that are part of the shared network. Devices are managed by the shared service provider’s server, and only registered devices are joined as members of the block network. Each device becomes a node in the network. The number of tokens remains consistent as per consensus among the groups that provide the service. Communication between nodes is performed using C-V2X (Cellular Vehicle to Everything). The initial token is issued by service provider groups, and tokens are transmitted after selecting random nodes in the block network. Node aggregates that received the token build a consensus network to create blocks among nodes that have tokens. Nodes that created blocks select random nodes in the block network and encrypt token information and own information with the corresponding node’s public key for transmission to the entire network. Nodes that have been deciphered with a private key acquire the authority to create the next block.

The block network random authentication node selection mechanism is divided into a registration module that registers mobility devices on the server, a token issue module that creates tokens based on the service provider’s consensus and provides them to the block network, a mobility transaction module that performs a consensus process to process mobility transaction, and a token transmission module that transmits tokens to other nodes.

### 3.1. Mobility Device Registration Module

In the mobility device registration module, the mobility service provider performs the device registration process via a server, which handles the provider’s mobility device. In other words, the block network’s node is limited only to mobility devices authenticated by the service provider. Information to recognize the mobility device is required to register the mobility device. Table 2 is a list of the data needed for the mobility device to be recognized in the proposed mechanism.

Authentication servers in block networks identify devices by utilizing the identification number created by the service provider and the MAD address of the IoT communication device included in the mobility device. The service provider creates device identification public information (*D_PI_*), which converts the identification number and MAC address to hash values, and transmits public device consensus data (*D_PA_*), including service type, service provider, and registration date, to the authentication server, taking part in building the mobility block network. Equation (1) shows the process of creating a hash value after performing a hash computation of the two identification information to identify the devices themselves. This is to keep too much information from being provided meaninglessly to other servers that do not look for direct identification information. Equation (2) means data structure to share information about mobility devices to utilize nodes on other servers.
(1)$$D_{PI}=Hash(N_{ID}+A_{M})$$
(2)$$D_{PA}=str(D_{PI}+S_{C}+S_{PV}+D)$$

Authentication servers in the block network that have public device consensus data check the public device consensus data received and transmit the number of mobility devices that have been requested to be added to the block network, the registration requested date, and information about the service provider to all authentication servers taking part in the block network. When identical information is shared across servers, a node creation authority whose number is the same as the aggregate of public device consensus data is added to the authentication server in the block network to which public device consensus data is transmitted. The authorized authentication server sets each mobility device as a node and builds a block network. Figure 2 shows the process of registering mobility devices into new nodes through consensus among authentication servers to have mobility devices participate in the block network in the mobility device registration module.

### 3.2. Token Issue Module

The token issue module is a module that issues tokens, including authentication information, and allows nodes that have a token on the block network to create a block. The token initially issued is created based on consensus among authentication servers participating in the block network. It includes the matching key to grant authority on token issuance.

Figure 3 shows the configuration of a token-created node transmitting a token to a random node belonging to the block network. Token transmission requires two levels of encryption. In the first one, the node that created the token encrypts the token with its own secret key to inform others that it has created a token. In the second one, encryption takes place with the target node’s public key to make sure token information is not caught by any node other than the target node that needs the token. As such, a node without a token can confirm the entire hash value, which is the correct answer to verify the authority of the block, and a question proving block creation authority, which is the question designed to verify authority. However, a secret key for block creation is needed to solve the question of proving block creation authority, and hence verification cannot be completed. A node with a token can verify a block by creating a secret key. Table 3 shows the information included in the token.

The token group matching key utilizes the identification information of the previous token group. When issuing a token for the first time, information from the token creation server that is in consensus with the token group matching key is used.

Figure 4 shows the process by which a block creates tokens. The token that is initially issued includes a matching key to build a network of nodes that have tokens. A secret key will be used as a verification means to acquire authority to create the next block, identify information from the server that creates the token, identify information about the node that will receive the token, and lastly, the value that is computed for the identification of information regarding the secret key and server, token order information, and node identification. Equation (4) shows the process of computing the hash value for all token values. When the block creation secret key included in the token is (*d*, *N*) and the block creation public key (*PU_BC_*) is (*e*, *N*), then the token that has not been encrypted is configured as shown in Equation (5).
(3)$$SK_{TG}=Hash(\sum_{i=0}^{n}TokenID\left[i\right])$$
(4)$$H_{TD}=Hash(SK_{TG}+PB_{BC}+CI_{TC}+CI_{TN}+R_{T})$$
(5)$$Token\;cert=(Encryption\;Key_{TG},\;Private\;Key\left(d,\;N\right),\;CI_{TC},\;CI_{TN},R_{T},H_{TD})$$

Figure 5 shows the formula process to broadcast a token encrypted by a block creation node to the block network. The token that has been created uses the public key (*e_t_*, *N_t_*) of the node set to receive the token to perform token encryption, as shown in Equation (6).
(6)$$Encryption\;Token={\left(Token\;cert\right)}^{et}\;mod\;N_{t}\;$$

Token is the created encrypted text. When it creates a hash and block of an encrypted token, it encrypts it with the secret key of the node that created the token, including the block creation authority verification question (*PB_BC_*) to verify authority to create. Equation (7) is the process of seeking a hash on the encrypted token, and Equation (8) is the process of how block creation authority is created by using the encrypted token. The token is eventually broadcast to the block network, as shown in Equation (9).
(7)$$T_{TO}=Hash(Encryption\;Token)$$
(8)$$PB_{BC}=(H_{TO}\;mod\;N)$$
(9)$$Broadcast\;Token={\left(Encryption\;Token+H_{TO}+PB_{BC}\right)}^{db}\;mod\;N_{b}$$

Figure 6 is the process of deciphering general nodes over which broadcast tokens have no authority over and tokens to node, which is the token’s subject. The server that created the token encrypts the token by using the public key of random nodes selected to transmit it to a random node in the block network, and the entire hash value, and the block creation authority verification question with its own secret key for broadcasting to the block network. Node checks that it has been sent from the server by deciphering the server’s public key and acquiring the encrypted token, entire hash value, and block creation authority verification question. Equation (10) shows the process of deciphering with a public key sent from the block to decipher the broadcast token.
(10)$$\begin{array}{cc} Plaintext\;Broadcast & ={\left(Broadcast\;Token\right)}^{eb}mod\;N_{b}\\ & =Encryption\;Token+H_{TO}+RB_{BC} \end{array}$$

The target node to which token should be sent uses its secret key to decipher token and acquire token group matching key in token and block creation secret key. Formula (11) is the process of the token target node deciphering an encrypted token with its own secret key (*d_t_*, *N_t_*) in order to decipher the node.
(11)$$Token={\left(Encryption\;Token\right)}^{dt}mod\;N_{t}\;=(Encryption\;key(SK_{TG},\;Pravate\;Key\left(d,\;N\right),\;CI_{CT},\;CI_{TN},\;R_{T},\;H_{TD})$$

Node creates a token group by using a matching key and collects transactions to create a block. Nodes then create blocks and perform verification with their secret keys. Nodes that failed to have a token use secret key included in block verification were used to decipher block creation authority verification questions and check if the results were identical with the entire hash value to confirm verification. Nodes that have created blocks afterwards create new tokens by selecting the secret key created by the node, which has a matching key and token that can be identically used among token groups, and a random node to which the token will be sent.

### 3.3. Mobility Transaction Module

The mobility transaction module is a series of processes for collecting transactions and providing mobility services to users. It performs a consensus process between the transaction’s data structure in the mobility node and the transaction itself. Mobility transaction data is created by the mobility device. It is executed when user information is forwarded to a mobility device upon user request. Mobility devices are composed of user information secured, additional time information, mobility device information, and regional information, which serve as one single transaction. The mobility transaction data structure can be described in Table 4.

Upon request, the users’ user mobility device fills out mobility transaction, data including user information received upon request. The transaction is encrypted as the mobility node’s secret key and broadcast to the block network. The broadcast transaction then performs verification via a token node. Figure 7 shows the execution process to verify the mobility transaction. All transactions are collected by token nodes, and blocks are created based on consensus between token nodes.

The collected transaction is collected by a token node, and hash computation is performed for each transaction to create a hash list (*L_TH_*). Equation (12) is the process whereby a hash list is created.
(12)$$L_{TH}=\sum_{i=0}^{n}Hash\left(Transaction\left[i\right]\right)$$

The first token node creates a transaction hash list (*L_TH_*), which is the result of executing a hash computation for the transaction collected. The transaction hash list that has been created is encrypted as the public key (*PU_TN_*) of the token node following in the next order and is encrypted as the token group matching key for broadcast to block the network. When the public key is (*e_nb_*, *N_nb_*) and the secret key is (*d_nb_*, *N_nb_*), the node that will receive the transaction hash list Equation (13) broadcasts it to the block network to show the process of deciphering the token group at a certain node. Equation (14) shows token group encryption when the secret key of the token group is a and SBox for encryption, and the matching table for deciphering is InvSBox.
(13)$$Next\;Token\;Node\;Encryption=({\left(L_{TH}\right)}^{enb}\;mod\;N_{nb})$$
(14)$$Token\;Group\;Encryption=a*SBox*({\left(L_{TH}\right)}^{enb}\;mod\;N_{nb})$$

The broadcast transaction list is deciphered first by the token group’s matching key and thus is shared only within the token group. Even within the group, the transaction hash list can be obtained only with a certain node’s secret key. Equation (15) is the process of a certain selected node executing deciphering via token group matching key. Equation (16) is the process of acquiring a transaction hash list by using its own secret key.
(15)$$1st\;Decryption=a*InvSBox*\left({\left(L_{TH}\right)}^{enb}\;mod\;N_{nb}\right)={\left(L_{TH}\right)}^{enb}\;mod\;N_{nb}$$
(16)$$2st\;Decryption={\left(L_{TH}\right)}^{dnb}\;mod\;N_{nb}=L_{TH}$$

The token node compares the broadcast transaction hash list with its own transaction hash list to perform verification with its own identification information for the same items. It then broadcasts the verified transaction hash list to the next token node, including the verification result of the transaction hash list. Equation (17) is the process of executing verification by comparing the transaction hash list with its own transaction hash list and performing renewal with the verification result including its own identification information. The renewed transaction list repeats the process from Equations (12)–(14) before sending to the next block.
(17)$$\begin{array}{c} Transaction\;List\\ =\left\{\begin{array}{c} Hash\left(Transaction\left[i\right]\right),\;CI_{TNx},\;if)Hash(Transaction[i])\in L_{THx+1}\\ Hash(Transaction\left[i\right]),\;else)Hash(Transaction[i]){\notin L}_{THx+1} \end{array}\right. \end{array}$$

Figure 8 shows the general consensus process of token nodes to perform transaction verification. A token node that has received a verified transaction hash list from all token nodes creates a transaction verification list (*L_TV_*), including the transaction list verified by all token nodes and the transaction’s hash verification list verified by more than half of the nodes. It is then encrypted with a token group matching key and broadcast to all networks. This is followed by a request for verification and its propagation to the incoming node. The token node that has been requested compares it with its own transaction list and sends it to all token groups when there are transactions with multiple verifications (*T_DV_*). Transactions not in the transaction list remove themselves from their own list and renew orders as per the list before propagating a request to perform verification at the node next in order. The first node requested for transaction verification by the Nth token node verifies its own block with the secret key in the authentication token and broadcasts to the node next in order. Token nodes that received blocks in consecutive order were then verified with their own secret keys. The Nth token node broadcasts to block network to block verification by all nodes when the number of verifiers is more than half of the entire token group.

### 3.4. Token Transmission Module

The token transmission module’s role is to create blocks, issue new tokens, and transmit them to the next block. The token group node uses the token’s block creation secret key to verify block verification in order to create a block. It then issues and attaches a token from among the next nodes that have been randomly selected. The newly-issued token creates a token group matching key, block creation secret key, token creation server identification information, target node identification information, and a token hash identical to the authentication token created by the server.

Token group matching key creates one matching key via token node group, and each token node uses the matching key created for its own token group matching key, which is then used as the matching key for the next node to forma token group. Since block creation public key pair is the solution to prove itself to create block, each token node creates an independent public key pair with different contents. Token creation server identification information enters the identification information of the node renewing token and is used to prove that node has been created by a random node receiving token. Target node identification information refers to the identification information of a random node, and token hash adds a hash value that performs computation on the token, just like the server. In short, the token node that created the block itself works as a single CA (Certificate Authority) and plays the role of a one-off certifying agency for the token.

Figure 9 is the token verification process. Random node that acquired token information confirms node that created token via token creation server identification information and encrypts its own identification information authentication token’s token hash with its public key for propagation (broadcasting). By using its secret key, the verification node identifies the authentication request made by the encrypted random node and encrypts the identification information of the verification node and verification findings with a random node public key for broadcast to the block network. A random node uses tokens as per verification results and creates a token node group.

Tokens that have been created perform encoding with the public key of a randomly selected node to prevent token details from being restored by a node other than the target node. It then performs encoding with the secret key of the node that creates a token to have the token-issuing entity verified on the block network. The encrypted token is then broadcast to the entire block network. Block network nodes that receive encrypted tokens utilize the public key to obtain encrypted token information, all hash value, and block creation authority verification questions, and use their own secret key to acquire token information. When token information is restored by its own secret key, it uses the token group matching key to create a token node group, and the token node creates a block by verifying transactions.

## 4. Discussion

In this paper, we review from the perspectives of stability, participability, and reliability to perform an analysis of the token-based meta-mobility block consensus network presented by the proposed mechanism.

### 4.1. Stability

During the process of creating blocks, the leakage of nodes with data generation authority plays a very important role in helping attackers select attack targets. Therefore, protecting nodes that hold authority will be a very important task [41].

#### 4.1.1. Challenges of Stability in Metamobility

In general, clients with the value of authority can be the target of attacks by malicious users who want to steal that value [42]. Therefore, nodes with authority must be safe from third-party attacks.

#### 4.1.2. Blockchain Contribution

The block interval varies depending on the consensus mechanism or usage of the blockchain. Bitcoin, a representative blockchain-based cryptocurrency, adjusts the creation difficulty and generates one block every 10 min on average [43]. In the case of Ethereum, blocks are created at intervals of about 1 to 20 s [44]. Therefore, creating blocks and sending tokens to other block creators means that the target node that is subject to attack continuously changes depending on the block interval.

#### 4.1.3. Challenges and Limitations

If the node that creates the block is a malicious user, it can have a significant and lasting impact on the block network [45]. In particular, if the process by which a token group node randomly selects the next token group node is not completely separated from the node’s users, only malicious nodes can be continuously selected, so the node selection process must be completely separated.

### 4.2. Participation Possibility

Common methodologies for creating blocks in the blockchain include PoS and PoW. In PoW, blocks are created by miners who solve problems through mining to create blocks, and in PoS, users with a large amount of tokens are more likely to create blocks [46]. Therefore, the PoW method requires a lot of resources to create a block, and the PoS method can form a centralized form based on stake [47,48].

#### 4.2.1. Challenges of Participation Possibility in Metamobility

The PoW method requires a lot of resources by participating in the block network and performing the process of finding the block hash to obtain rewards, so it is difficult to apply to IoT, which cannot handle many resources. Since the PoS method creates blocks based on stake, it can cause monopolies [49]. In particular, in the mobility IoT environment where miniaturization is required, it is necessary to build a block network using low-performance resources [50].

#### 4.2.2. Blockchain Contribution

Distributing block creation rights is a very important factor in maintaining the block network safely. In the PoW method, the safety of the network is strengthened by spending time and money acquiring the right to create blocks according to the workload, while the PoS method increases the cost of malicious behavior by providing different possibilities for creating blocks depending on the stake [51]. In the proposed mechanism, the authority to create a block is created only by the consensus of a group of token nodes with tokens based on the initially generated token, and the node that acquires the token is selected randomly, providing the possibility to all nodes.

#### 4.2.3. Challenges and Limitations

The PoW method enhances safety by requiring a high computational effort for malicious actions by incurring a cost to create a block, and the PoS method guarantees that nodes with a large stake will take losses and will not commit malicious actions on the network. Strengthening safety through assumptions [52]. However, since the proposed mechanism randomly selects the node that will receive the token, there is a possibility that a malicious node may be selected. Therefore, research is required to strengthen the safety of the block network in cases where the majority of the selected token group are malicious nodes.

### 4.3. Reliability

In a blockchain, it is an important element to prove that the party creating the block has the authority to create the block. The PoW method and the PoS method are methods that use work and stake as means of proof, respectively [24].

#### 4.3.1. Challenges of Reliability in Metamobility

The mobility environment provides services through various communication methods, such as DSRC (Dedicated Short Range Communication) or LTE [53]. In particular, due to the nature of the mobility environment, there is a high possibility that it is not connected to a power source, and it is difficult to apply high computing power to sustain the service for a long time [54]. Therefore, a low-performance blockchain that can be applied even at low performance is required.

#### 4.3.2. Blockchain Contribution

In the process of creating blocks, the PoW method requires high performance to obtain block creation rights, so there are many difficulties in applying it [55]. The PoS method has the disadvantage that the entity that creates blocks according to the stake may gradually become entrenched in the nodes that have increased the stake over a long period of time [56]. The proposed mechanism performs the process of proving authority using the private key of the public key pair contained in the token. Additionally, since the private key changes every round of creating a block, no one other than the node that created the token and the node that received the token can obtain the public key pair, making token-based proof possible.

#### 4.3.3. Challenges and Limitations

Authentication based on tokens can be obtained from two nodes, with the generator generating the token and the user acquiring the token. Therefore, tokens generated by malicious users can be recycled by malicious nodes. Therefore, research is required on the token creation process and ways to restrict access to the generated tokens.

## 5. Results

The random token-based selection method proposed in this paper allows only nodes that have tokens to take part in the group for block creation and creates consensus blocks in the token group network. As for nodes authorized to create blocks, random nodes on the block network are selected, and therefore all nodes have a chance to take part in block creation. In addition, it does not require high performance since there is no computation process to acquire block creation authority. However, nodes authorized to create blocks maintain a consistent number, and there is a risk of block forgery/counterfeit if more than half of the selected nodes have malign intentions. In addition, an issue of redundancy exists where more than two token nodes can set one single node into the next token node.

In this paper, in order to analyze the differences between other consensus mechanisms and the proposed mechanism, the method of selecting the entity that generates the block in each consensus mechanism, whether mining is necessary, the computing power required for the operation of the consensus mechanism, whether representative nodes are selected, and whether or not they are anonymous are discussed. The analysis was conducted focusing on five items.

In the method of selecting the entity that creates the block, the general blockchain consensus mechanism focuses on work and stake. First, the work-based consensus mechanism is a method in which the subject who has performed the most of the process of finding a nonce value that generates a hash value smaller than the target value for a specific block participates in block creation. The stake-based consensus mechanism performs block creation based on resource ownership on the block network. Therefore, the work-based PoW method performs competition to find more nonces to generate blocks. The stake-based PoS or DPoS method performs participation using resource stakes, so as resources accumulate, it is very difficult for users who join the block network late to participate in block creation. The proposed mechanism presented the possibility of creating blocks for all users participating in the block network by assigning block participation rights based on the tokens issued for block creation.

Next, we look at the necessity of the mining process from the perspective of PoW, PoS, DPoS, and proposed mechanisms. In general, the concept of mining refers to the process of finding a new hash by changing the nonce value to find a hash value smaller than the hash value of a specific block in order to create a block in the PoW method. Since the PoS method participates in block creation according to one’s stake, the process of finding the nonce value is not required. In the proposed mechanism, when creating a block, a node with a token verifies the block using the secret key contained in the token. Therefore, it does not require proof of work or proof of stake for block creation.

In a general mobility IoT environment, computing power does not require high performance, and therefore, in order to apply blockchain, a distributed network environment, in a mobility IoT environment, lightweighting is required. The PoW method requires high computing performance because it increases the likelihood that high-performance nodes will generate blocks during the mining process. The PoS method does not require competition because nodes use their stakes to prove blocks and can be applied with relatively low computing power. The proposal mechanism uses a medium called a token to allocate block creation rights, so there is no element of competition and it does not require high computing power.

In order to perform faster processing in the block network, there is a node called a representative node that performs proof on behalf of the node. The PoW method is a method in which individual nodes compete, and the PoS method is a method in which each node uses its own stake to prove, so there is no representative node. In the DPoS method, proof of one’s stake is performed by delegating it to a representative node. In the proposed mechanism, the node with the token becomes the representative node that creates the block, so there is a representative node that performs the creation of the block on its behalf.

Anonymity means that nodes must prove themselves before creating a block. In the PoS and DPoS methods, nodes or representative nodes use their stake to prove blocks, so each node must be identified during the stake proof process for the block. However, since the PoW method changes the nonce and searches for the hash, it assigns block creation rights according to the amount of resources, so it is not possible to recognize nodes with block creation rights in advance. In the proposed mechanism, before transmitting the token for assigning block creation authority, it is first encrypted and broadcast with the public key of the recipient. Since the encrypted token can only be decrypted using the secret key, the token cannot be recognized by anyone other than the node that will receive it, and the block-generating node cannot be recognized.

Table 5 shows the difference in consensus between the existing consensus mechanism and the proposed mechanism. PoW needs to perform the biggest workload for block creation authority. IAn PoS node with the highest stakes has a higher chance of acquiring block creation authority. In DPoS, stakes may play a key role in participating in block creation, but it is different in that stakes are used to select representatives. The proposed mechanism utilizes token broadcast to the randomly selected node in selecting a representative. High-performance mining is required only in PoW and not in PoS, DPoS, or the proposed mechanism. The node itself is the verifier in PoW and PoS in that a key node is selected to create block, while a key node is selected in DPoS, and the proposed mechanism is determined by a vote based on each stake and a random selection. PoW proof can be executed anonymously, but node information is required for PoS and DPoS when selecting stakes or representatives. In the proposed mechanism, a token group matching key is used to let nodes with tokens only take part in the network. As such, a node without a token cannot gain token group information.

## 6. Conclusions and Future Works

Advances in small IoT devices and growing demand for electric power devices are bringing major changes in the size of mobility services. In vehicles, physical power devices for control as an independent entity are giving way to sensor-controlled vehicle functions applied with intelligent vehicle technologies to provide user convenience. Even small mobility devices such as kick-boards and bicycles are seeing a shift from human-driven power to power devices based on electric batteries for mobility service. This suggests that mobility services are relying more on intelligent mobility devices, and control is increasingly based on sensors.

Intelligent control based on sensors can provide convenience to mobility service users, but it runs the risk of ill intentions by forging data or maliciously counterfeiting mobility service data to harm the mobility big data environment. Attacks when controlling intelligent vehicles via sensor communication or communicating with shared mobility services need to be prevented. At the same time, a verification means to build reliable mutual communication, and this is where block chain is coming into play as a solution.

This paper proposed a mechanism that randomly selects token nodes by using a token, which creates blocks in the block network that have been built by a device authenticated by a restricted authentication server in order to apply block chaining in the mobility environment. By selecting a random node as the node to create a block, block creation fixated by a certain node can be prevented. In addition, token node information is not provided to another node since a node that has been transmitted with a token can decipher token information only with its secret key. However, in the proposed mechanism, tokens are sent to random nodes to create blocks, but a fatal problem may occur if the node that receives the token is a malicious node. In this case, the node can arbitrarily transmit the token to another group of malicious nodes. In future research, we plan to conduct research to prevent malicious users from randomly selecting other nodes during the process of randomly selecting nodes.

## Figures and Tables

**Figure 1 sensors-23-08259-f001:**
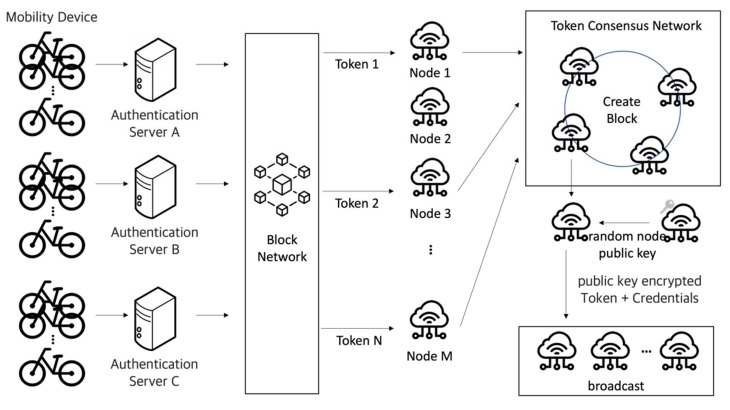
Conceptual drawing of a block network random authentication node selection mechanism.

**Figure 2 sensors-23-08259-f002:**
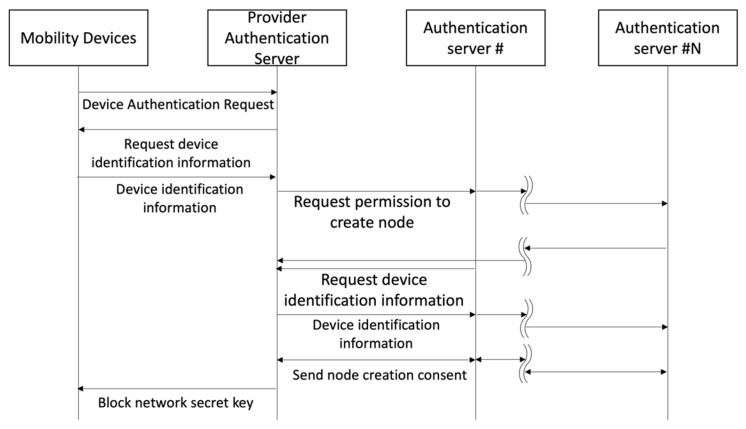
Mobility device node registration process.

**Figure 3 sensors-23-08259-f003:**
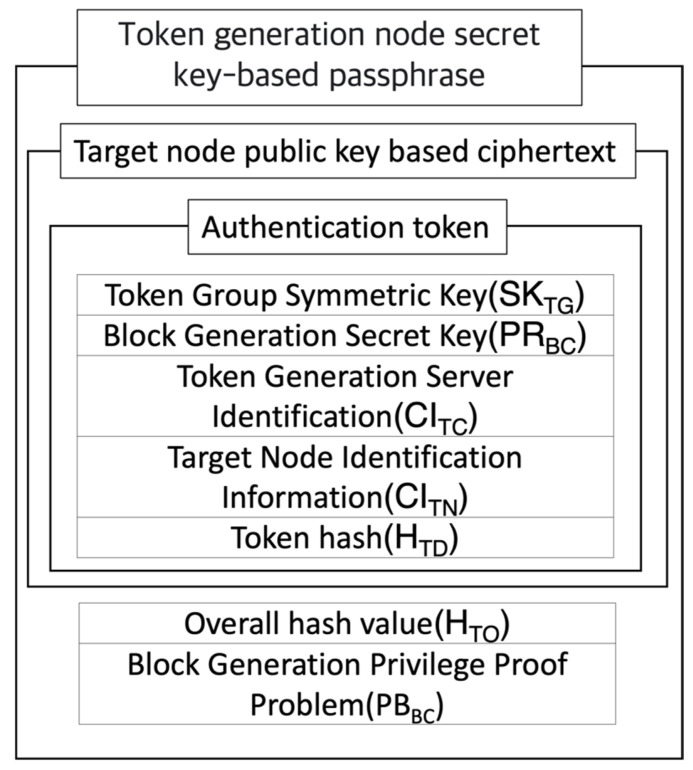
Token transmission configuration.

**Figure 4 sensors-23-08259-f004:**
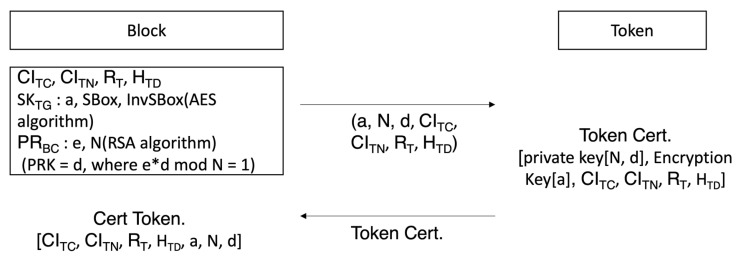
Token creation process.

**Figure 5 sensors-23-08259-f005:**
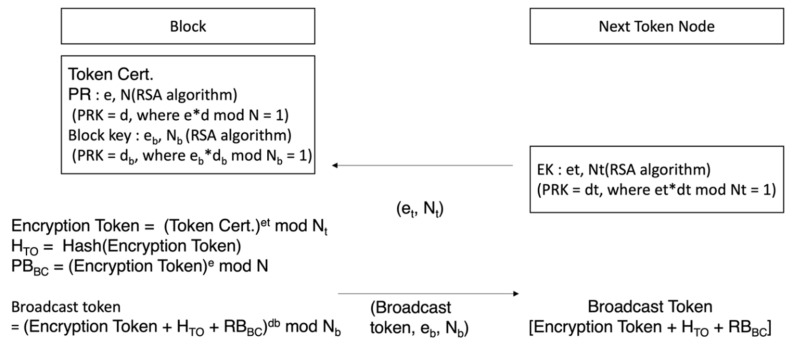
Process of creating questions to prove block creation authority.

**Figure 6 sensors-23-08259-f006:**
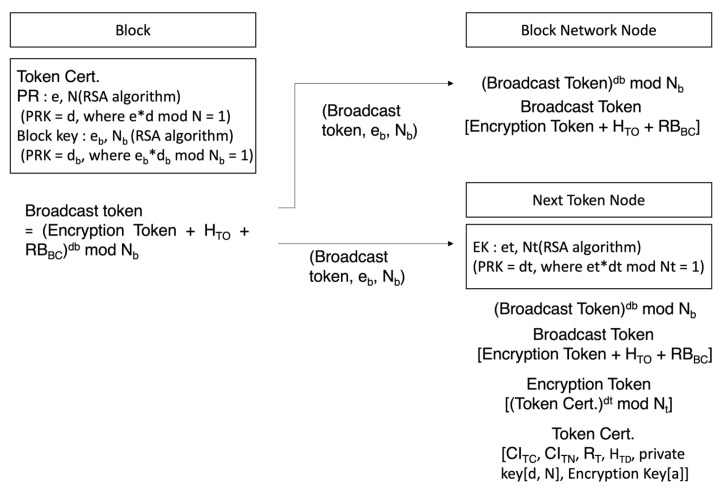
Broadcast token deciphering process of a general node and a node subject to a token.

**Figure 7 sensors-23-08259-f007:**
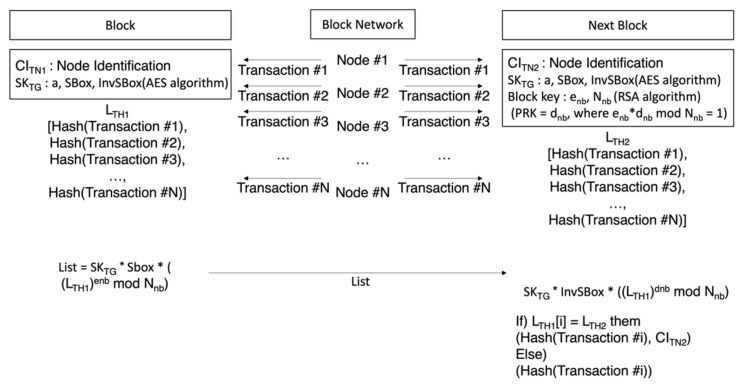
Mobility transaction verification data broadcast process.

**Figure 8 sensors-23-08259-f008:**
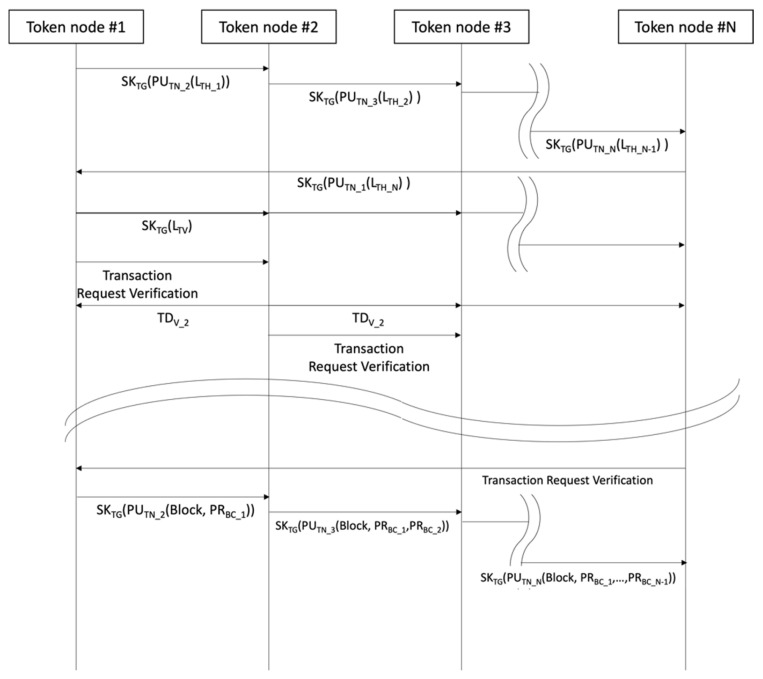
Transaction verification process by token node.

**Figure 9 sensors-23-08259-f009:**
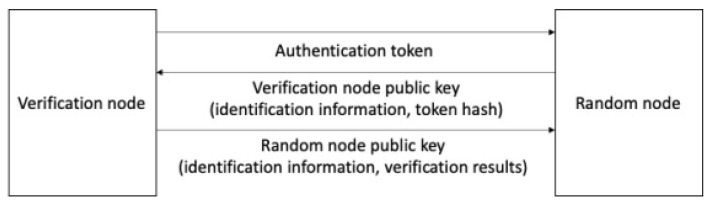
Token authentication process.

**Table 1 sensors-23-08259-t001:** Comparison of IoT networks.

Category	LoRa	NB-IoT	eMTC
Coverage	~10 km	~15 km	~11 km
Bandwidth	920–925 MHz (Republic of Korea)	180 kHz	1.4 MHz
Transmission speed	10 kbps	~250 kbps	~1 Mbps
Battery life	~10 years	~10 years	~10 years

**Table 2 sensors-23-08259-t002:** Classification of mobility device data.

Data	Description
Identification no. (NID)	Device identification information created by the service provider
MAC address (AM)	Physical address of mobility IoT network device
Service type (SC)	Types of mobility services provided
Service provider (SPV)	Service provider’s identification information
Registration date (D)	Mobility information registration date

**Table 3 sensors-23-08259-t003:** Token configuration.

Category	Description
Token group matching key (*SK_TG_*)	Code to build a network between nodes with a token
Block creation secret key (*PR_BC_*)	Answer to prove block creation authority
Token creation server identification information (*CI_TC_*)	Identification information of the server that created a token
Target node identification information (*CI_TN_*)	Identification information of the node to receive the token
Token order information (*R_T_*)	Information marking token order
Token hash (*H_TD_*)	Hash computation value for all token characteristics

**Table 4 sensors-23-08259-t004:** Mobility transaction data structure.

Data	Transaction	Description
User information	User identification information	Authentication information to identify a hash-computed user
Time information	Time service starts	Time service starts as per request
Time information	Time service ends	Time service ends as per request
Regional information	Administrative district code	Administrative district identification code where the mobility node is located
Regional information	GPS information	GPS information of the mobility node
Mobility device information	Node identification information	Mobility device identification information
Mobility device information	Distance information	Specific users’ distance information

**Table 5 sensors-23-08259-t005:** Comparative analysis of existing consensus mechanisms and proposed mechanisms.

	PoW	PoS	DPoS	Proposed Mechanism
Block creation authority	Work	Stake	Stake	Token
Mining	O	X	X	X
Required computation power	High	Low	Low	Low
Representative node	X	X	O	O
Anonymity	O	X	X	O

## Data Availability

Data sharing not applicable.

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
