# Peer review of "Proposal of a Token-Based Node Selection Mechanism for Node Distribution of Mobility IoT Blockchain Nodes"

_sensors, 2023, doi:10.3390/s23198259_

Round 1

Reviewer 1 Report

I thank the authors for reviewing their study. The manuscript is original and is interesting because it proposes a random node selection mechanism where tokens are provided to create blocks by strengthening data integrity. Minor revisions are required:

1.            The title of the manuscript is very articulate. I suggest adding words such as blockchain, internet of things, token. Also, it should briefly summarize the contents of the study;

2.            The objective of the research should be more clearly expressed in the abstract;

3.            Add at the end of the Introduction what is the structure of the paper and its subsequent sections.

Good luck!

Minor editing of English language required

Author Response

We are very grateful for your recommendations for corrections to improve the manuscript.

Modifications were made based on the reviewer's recommendations, and detailed corrections were compiled and attached as a word file.

Thank you again for your help. If you have any additional recommendations for improving the manuscript, we will do our best to revise it.

Thank you.

Reviewer 2 Report

This is an interesting manuscript, in which the authors have proposed a method for random certificate node selection in blockchain. Although the topic of the manuscript is interesting and worthy of investigation, there are several issues that should be resolved by the authors.

1.       The comparative analysis presented in Section 5 is interesting. However, it is recommended that the authors elaborate this Section, by discussing the validation and comparison methodologies. Concretely, the discussion of metrics and/or the monitoring of Key Performance Indicators would be useful in order to assess the proposed method.

2.       In the presented manuscript, the authors present an interesting method. However, in order to improve its impact, it is recommended to revise the Conclusions Section, in order to provide an adequate Outlook for the manuscript. Concretely, the authors are asked to provide explicit future research directions based on the current implications and limitations.

3.       In the first paragraph of the introduction section too many different concepts are introduced. Therefore, please simplify the text and maintain only the necessary components.

4.       The architecture presented in Figure 1 is interesting. However, it is recommended that the authors elaborate it by presenting in greater detail the information flow between the individual modules, as well as indicate the communication protocols implemented.

5.       The technical presentation of the manuscript is limited to a theoretical level, meaning that the authors have neither included a case study, nor discussed any numerical results. Therefore, please revise the manuscript accordingly, in order to improve the impact of the presented work.

6.       Regarding the discussion of the consensus mechanisms in the blockchain, it is recommended that the authors proceed with a detailed comparative analysis. More information can be drawn from the following publication:

a.       Mourtzis, D.; Angelopoulos, J.; Panopoulos, N. Blockchain Integration in the Era of Industrial Metaverse. Appl. Sci. 2023, 13, 1353. DOI: https://doi.org/10.3390/app13031353

7.       Please avoid having single-sentence paragraphs in the text.

8.       In the current version of the manuscript, in the first sentence of the last paragraph in the Introduction Section, the authors state in their own words that “Mechanism proposed in this thesis provides tokens…” This is not a thesis. Rather this is an original research contribution. Therefore, please revise accordingly.

9.       The quality of the diagrams presented in all the figures should be further elaborated in order to improve their readability.

10.   At the end of the Introduction Section, it is recommended to add a short paragraph in order to describe the structure of the manuscript.

An additional proof-read run by a native English speaker is recommended in order to eliminate any grammar/syntax errors.

Author Response

We are very grateful for your recommendations for corrections to improve the manuscript.

Modifications were made based on the reviewer's recommendations, and detailed corrections were compiled and attached as a word file.

Thank you again for your help. If you have any additional recommendations for improving the manuscript, we will do our best to revise it.

thank you
